# PRECONDITIONER ON MATRIX LIE GROUP FOR SGD

**Xi-Lin Li**
GMEMS Technologies and Spectimbre
366 Fairview Way, Milpitas, CA 95035
`lixilinx@gmail.com`

## ABSTRACT

We study two types of preconditioners and preconditioned stochastic gradient descent (SGD) methods in a unified framework. We call the first one the Newton type due to its close relationship to the Newton method, and the second one the Fisher type as its preconditioner is closely related to the inverse of Fisher information matrix. Both preconditioners can be derived from one framework, and efficiently estimated on any matrix Lie groups designated by the user using natural or relative gradient descent minimizing certain preconditioner estimation criteria. Many existing preconditioners and methods, e.g., RMSProp, Adam, KFAC, equilibrated SGD, batch normalization, etc., are special cases of or closely related to either the Newton type or the Fisher type ones. Experimental results on relatively large scale machine learning problems are reported for performance study.

## 1 INTRODUCTION

This paper investigates the use of preconditioner for accelerating gradient descent, especially in large scale machine learning problems. Stochastic gradient descent (SGD) and its variations, e.g., momentum (Rumelhart et al., 1986; Nesterov, 1983), RMSProp and Adagrad (John et al., 2011), Adam (Kingma & Ba, 2015), etc., are popular choices due to their simplicity and wide applicability. These simple methods do not use well normalized step size, could converge slow, and might involve more controlling parameters requiring fine tweaking. Convex optimization is a well studied field (Boyd & Vandenberghe, 2004). Many off-the-shelf methods there, e.g., (nonlinear) conjugate gradient descent, quasi-Newton methods, Hessian-free optimizations, etc., can be applied to small and middle scale machine learning problems without much modifications. However, these convex optimization methods may have difficulty in handling gradient noise and scaling up to problems with hundreds of millions of free parameters. For a large family of machine learning problems, natural gradient with the Fisher information metric is equivalent to a preconditioned gradient using inverse of the Fisher information matrix as the preconditioner (Amari, 1998). Natural gradient and its variations, e.g., Kronecker-factored approximate curvature (KFAC) (Martens & Grosse, 2015) and the one in (Povey et al., 2015), all use such preconditioners. Other less popular choices are the equilibrated preconditioner (Dauphin et al., 2015) and the one proposed in (Li, 2018). Momentum or the heavy-ball method provides another independent way to accelerate converge (Nesterov, 1983; Rumelhart et al., 1986). Furthermore, momentum and preconditioner can be combined to further accelerate convergence as shown in Adam (Kingma & Ba, 2015).

This paper groups the above mentioned preconditioners and preconditioned SGD methods into two classes, the Newton type and the Fisher type. The Newton type is closely related to the Newton method, and is suitable for general purpose optimizations. The Fisher type preconditioner relates to the inverse of Fisher information matrix, and is limited to a large subclass of stochastic optimization problems where the Fish information metric can be well defined. Both preconditioners can be derived from one framework, and estimated on any matrix Lie groups designated by the user with almost the same natural or relative gradient descent methods minimizing specific preconditioner estimation criteria.

## 2 BACKGROUND

### 2.1 NOTATIONS

We consider the minimization of cost function

$$f(\boldsymbol{\theta}) = E_z[\ell(\boldsymbol{\theta}, \boldsymbol{z})] \qquad (1)$$

where $E_z$ takes expectation over random variable $\boldsymbol{z}$, $\ell$ is a loss function, and $\boldsymbol{\theta}$ is the model parameter vector to be optimized. For example, in a classification problem, $\ell$ could be the cross entropy loss, $\boldsymbol{z}$ is a pair of input feature vector and class label, vector $\boldsymbol{\theta}$ consists of all the trainable parameters in the classification model, and $E_z$ takes average over all samples from the training data set. By assuming second order differentiable model and loss, we could approximate $\ell(\boldsymbol{\theta}, \boldsymbol{z})$ as a quadratic function of $\boldsymbol{\theta}$ within a trust region around $\boldsymbol{\theta}$, i.e., $\ell(\boldsymbol{\theta}, \boldsymbol{z}) = \boldsymbol{b}_z^T \boldsymbol{\theta} + 0.5\boldsymbol{\theta}^T \boldsymbol{H}_z \boldsymbol{\theta} + a_z$, where $a_z$ is the sum of higher order approximation errors and constant term independent of $\boldsymbol{\theta}$, $\boldsymbol{H}_z$ is a symmetric matrix, and subscript $z$ in $\boldsymbol{b}_z$, $\boldsymbol{H}_z$ and $a_z$ reminds us that these three terms depend on $\boldsymbol{z}$. Clearly, these three terms depend on $\boldsymbol{\theta}$ as well, although we do not explicitly show this dependence to simplify our notations since we just consider parameter updates in the same trust region. Now, we may rewrite (1) as $f(\boldsymbol{\theta}) = \boldsymbol{b}^T \boldsymbol{\theta} + 0.5\boldsymbol{\theta}^T \boldsymbol{H} \boldsymbol{\theta} + a$, where $\boldsymbol{b} = E_z[\boldsymbol{b}_z]$, $\boldsymbol{H} = E_z[\boldsymbol{H}_z]$, and $a = E_z[a_z]$. We do not impose any assumption, e.g., positive definiteness, on $\boldsymbol{H}$ except for being symmetric. Thus, the quadratic surface in the trust region could be non-convex. To simplify our notations, we no longer consider the higher order approximation errors included in $a$, and simply assume that $f(\boldsymbol{\theta})$ is a quadratic function of $\boldsymbol{\theta}$ in the trust region.

### 2.2 PRECONDITIONER

Let us consider a certain iteration. Preconditioned SGD updates $\boldsymbol{\theta}$ as

$$\boldsymbol{\theta} \leftarrow \boldsymbol{\theta} - \mu \boldsymbol{P} \, \partial \hat{f}(\boldsymbol{\theta})/\partial \boldsymbol{\theta} \qquad (2)$$

where $\mu > 0$ is the step size, $\hat{f}(\boldsymbol{\theta})$ is an estimate of $f(\boldsymbol{\theta})$ obtained by replacing expectation with sample average, and positive definite matrix $\boldsymbol{P}$ could be a fixed or adaptive preconditioner. By letting $\boldsymbol{\theta}' = \boldsymbol{P}^{-0.5}\boldsymbol{\theta}$, we can rewrite (2) as

$$\boldsymbol{\theta}' \leftarrow \boldsymbol{\theta}' - \mu \, \partial \hat{f}(\boldsymbol{\theta})/\partial \boldsymbol{\theta}' \qquad (3)$$

where $\boldsymbol{P}^{-0.5}$ denotes the principal square root of $\boldsymbol{P}$. Hence, (3) suggests that preconditioned SGD is equivalent to SGD in a transformed parameter domain. Within the considered trust region, let us write the stochastic gradient, $\partial \hat{f}(\boldsymbol{\theta})/\partial \boldsymbol{\theta}$, explicitly as

$$\partial \hat{f}(\boldsymbol{\theta})/\partial \boldsymbol{\theta} = \hat{\boldsymbol{H}}\boldsymbol{\theta} + \hat{\boldsymbol{b}} \qquad (4)$$

where $\hat{\boldsymbol{H}}$ and $\hat{\boldsymbol{b}}$ are estimates of $\boldsymbol{H}$ and $\boldsymbol{b}$, respectively. Combining (4) and (2) gives the following linear system

$$\boldsymbol{\theta} \leftarrow (\boldsymbol{I} - \mu\boldsymbol{P}\hat{\boldsymbol{H}})\boldsymbol{\theta} - \mu\boldsymbol{P}\hat{\boldsymbol{b}} \qquad (5)$$

for updating $\boldsymbol{\theta}$ within the assumed trust region, where $\boldsymbol{I}$ is the identity matrix. A properly determined $\boldsymbol{P}$ could significantly accelerate convergence of the locally linear system in (5).

We review a few facts shown in (Li, 2018) before introducing our main contributions. Let $\delta\boldsymbol{\theta}$ be a random perturbation of $\boldsymbol{\theta}$, and be small enough such that $\boldsymbol{\theta} + \delta\boldsymbol{\theta}$ still resides in the same trust region. Then, (4) suggests the following resultant perturbation of stochastic gradient,

$$\delta\hat{\boldsymbol{g}} \stackrel{\text{def}}{=} \partial\hat{f}(\boldsymbol{\theta} + \delta\boldsymbol{\theta})/\partial\boldsymbol{\theta} - \partial\hat{f}(\boldsymbol{\theta})/\partial\boldsymbol{\theta} = \hat{\boldsymbol{H}}\delta\boldsymbol{\theta} = \boldsymbol{H}\delta\boldsymbol{\theta} + \boldsymbol{\varepsilon} \qquad (6)$$

where $\boldsymbol{\varepsilon}$ accounts for the error due to replacing $\hat{\boldsymbol{H}}$ with $\boldsymbol{H}$. Note that by definition, $\delta\hat{\boldsymbol{g}}$ is a random vector dependent on both $\boldsymbol{z}$ and $\delta\boldsymbol{\theta}$. The preconditioner in (Li, 2018) is pursued by minimizing criterion

$$c(\boldsymbol{P}) = E_{z,\delta\theta}[\delta\hat{\boldsymbol{g}}^T\boldsymbol{P}\delta\hat{\boldsymbol{g}} + \delta\boldsymbol{\theta}^T\boldsymbol{P}^{-1}\delta\boldsymbol{\theta}] \qquad (7)$$

where subscript $\delta\theta$ in $E_{z,\delta\theta}$ denotes taking expectation over $\delta\boldsymbol{\theta}$. Under mild conditions, criterion (7) determines a unique positive definite $\boldsymbol{P}$, which is optimal in the sense that it preconditions the stochastic gradient such that

$$\boldsymbol{P}E_{z,\delta\theta}[\delta\hat{\boldsymbol{g}}\delta\hat{\boldsymbol{g}}^T]\boldsymbol{P} = E_{\delta\theta}[\delta\boldsymbol{\theta}\delta\boldsymbol{\theta}^T] \qquad (8)$$

which is comparable to relationship $\boldsymbol{H}^{-1}\delta\boldsymbol{g}\delta\boldsymbol{g}^T\boldsymbol{H}^{-1} = \delta\boldsymbol{\theta}\delta\boldsymbol{\theta}^T$, where $\delta\boldsymbol{g} = \boldsymbol{H}\delta\boldsymbol{\theta}$ is the perturbation of noiseless gradient, and we assume that $\boldsymbol{H}$ is invertible, but not necessarily positive definite. Clearly, this preconditioner is comparable to $\boldsymbol{H}^{-1}$. It perfectly preconditions the gradient such that the amplitudes of parameter perturbations match that of the associated preconditioned gradient, regardless of the amount of gradient noise. Naturally, preconditioned SGD with this preconditioner inherits the scale-invariance property of the Newton method.

Note that in the presence of gradient noise, the optimal $\boldsymbol{P}$ and $\boldsymbol{P}^{-1}$ given by (8) are not unbiased estimates of $\boldsymbol{H}^{-1}$ and $\boldsymbol{H}$, respectively. Actually, even if $\boldsymbol{H}$ is positive definite and available, $\boldsymbol{H}^{-1}$ may not always be a good preconditioner when $\boldsymbol{H}$ is ill-conditioned since it could significantly amplify the gradient noise along the directions of the eigenvectors of $\boldsymbol{H}$ associated with small eigenvalues, and lead to divergence. More specifically, it is shown in (Li, 2018) that $\boldsymbol{H}^{-1}E_{z,\delta\theta}[\delta\hat{\boldsymbol{g}}\delta\hat{\boldsymbol{g}}^T]\boldsymbol{H}^{-1} \geq E_{\delta\theta}[\delta\boldsymbol{\theta}\delta\boldsymbol{\theta}^T]$, where $\boldsymbol{A} \geq \boldsymbol{B}$ means that $\boldsymbol{A} - \boldsymbol{B}$ is nonnegative definite.

## 3 TWO PRECONDITIONER ESTIMATION CRITERIA

### 3.1 THE NEWTON TYPE CRITERION

Preconditioner estimation criterion (7) requires $\delta\boldsymbol{\theta}$ to be small enough such that $\boldsymbol{\theta}$ and $\boldsymbol{\theta} + \delta\boldsymbol{\theta}$ reside in the same trust region. In practice, numerical error might be an issue when handling small numbers with floating point arithmetic. This concern becomes more grave with the popularity of single and even half precision math in large scale neural network training. Luckily, (6) relates $\delta\hat{\boldsymbol{g}}$ to the Hessian-vector product, which can be efficiently evaluated with automatic differentiation software tools. Let $\boldsymbol{v}$ be a random vector with the same dimension as $\boldsymbol{\theta}$. Then, (4) suggests the following method for Hessian-vector product evaluation,

$$\frac{\partial}{\partial\boldsymbol{\theta}}\left\{\left[\partial\hat{f}(\boldsymbol{\theta})/\partial\boldsymbol{\theta}\right]^T\boldsymbol{v}\right\} = \frac{\partial^2\hat{f}(\boldsymbol{\theta})}{\partial\boldsymbol{\theta}\partial\boldsymbol{\theta}^T}\boldsymbol{v} = \hat{\boldsymbol{H}}\boldsymbol{v} \tag{9}$$

Now, replacing $(\delta\boldsymbol{\theta}, \delta\hat{\boldsymbol{g}})$ in (7) with $(\boldsymbol{v}, \hat{\boldsymbol{H}}\boldsymbol{v})$ leads to our following new preconditioner estimation criterion,

$$c_n(\boldsymbol{P}) = E_{z,v}[\boldsymbol{v}^T\hat{\boldsymbol{H}}\boldsymbol{P}\hat{\boldsymbol{H}}\boldsymbol{v} + \boldsymbol{v}^T\boldsymbol{P}^{-1}\boldsymbol{v}] \tag{10}$$

where the subscript $v$ in $E_{z,v}$ suggests taking expectation over $\boldsymbol{v}$. We no longer have the need to assume $\boldsymbol{v}$ to be an arbitrarily small vector. It is important to note that this criterion only requires the Hessian-vector product. The Hessian itself is not of interest. We call (10) the Newton type preconditioner estimation criterion as the resultant preconditioned SGD method is closely related to the Newton method.

### 3.2 THE FISHER TYPE CRITERION

We consider the machine learning problems where the *empirical* Fisher information matrix can be well defined by $\boldsymbol{F} = E_z\left[\frac{\partial\ell(\boldsymbol{\theta},\boldsymbol{z})}{\partial\boldsymbol{\theta}}\left(\frac{\partial\ell(\boldsymbol{\theta},\boldsymbol{z})}{\partial\boldsymbol{\theta}}\right)^T\right]$. Replacing $(\delta\boldsymbol{\theta}, \delta\hat{\boldsymbol{g}})$ in (7) with $(\boldsymbol{v}, \hat{\boldsymbol{g}} + \lambda\boldsymbol{v})$ leads to criterion

$$c_f(\boldsymbol{P}) = E_{z,v}[(\hat{\boldsymbol{g}} + \lambda\boldsymbol{v})^T\boldsymbol{P}(\hat{\boldsymbol{g}} + \lambda\boldsymbol{v}) + \boldsymbol{v}^T\boldsymbol{P}^{-1}\boldsymbol{v}] \tag{11}$$

where $\hat{\boldsymbol{g}} = \partial\hat{f}(\boldsymbol{\theta})/\partial\boldsymbol{\theta}$ is a shorthand for stochastic gradient, and $\lambda \geq 0$ is a damping factor. Clearly, $\boldsymbol{v}$ is independent of $\hat{\boldsymbol{g}}$. Let us further assume that $\boldsymbol{v}$ is drawn from standard multivariate normal distribution $\mathcal{N}(\boldsymbol{0}, \boldsymbol{I})$, i.e., $E_v[\boldsymbol{v}] = \boldsymbol{0}$ and $E_v[\boldsymbol{v}\boldsymbol{v}^T] = \boldsymbol{I}$. Then, we could simplify $c_f(\boldsymbol{P})$ as

$$c_f(\boldsymbol{P}) = \text{tr}\{\boldsymbol{P}E_{z,v}[(\hat{\boldsymbol{g}} + \lambda\boldsymbol{v})(\hat{\boldsymbol{g}} + \lambda\boldsymbol{v})^T] + \boldsymbol{P}^{-1}E_v[\boldsymbol{v}\boldsymbol{v}^T]\}$$
$$= \text{tr}\{\boldsymbol{P}[E_z[\hat{\boldsymbol{g}}\hat{\boldsymbol{g}}^T] + \lambda^2\boldsymbol{I}] + \boldsymbol{P}^{-1}\} \tag{12}$$

By letting the derivative of $c_f(\boldsymbol{P})$ with respect to $\boldsymbol{P}$ be zero, the optimal positive definite solution for $c_f(\boldsymbol{P})$ is readily shown to be

$$\boldsymbol{P} = \{E_z[\hat{\boldsymbol{g}}\hat{\boldsymbol{g}}^T] + \lambda^2\boldsymbol{I}\}^{-0.5} \tag{13}$$

When $\hat{\boldsymbol{g}}$ is a gradient estimation obtained by taking average over $B$ independent samples, $E_z[\hat{\boldsymbol{g}}\hat{\boldsymbol{g}}^T]$ is related to the Fisher information matrix by

$$E_z[\hat{\boldsymbol{g}}\hat{\boldsymbol{g}}^T] = \boldsymbol{F}/B + (B-1)\boldsymbol{g}\boldsymbol{g}^T/B \tag{14}$$

We call this preconditioner the Fisher type one due to its close relationship to the Fisher information matrix. One can easily modify this preconditioner to obtain an unbiased estimation of $\boldsymbol{F}^{-1}$. Let $\boldsymbol{s}$ be an exponential moving average of $\hat{\boldsymbol{g}}$. Then, after replacing the $\hat{\boldsymbol{g}}$ in (13) with $\hat{\boldsymbol{g}} - \boldsymbol{s} + \boldsymbol{s}/\sqrt{B}$ and setting $\lambda = 0$, $\boldsymbol{P}^2/B$ will be an unbiased estimation of $\boldsymbol{F}^{-1}$. Generally, it might be acceptable to keep the bias term, $(B-1)\boldsymbol{g}\boldsymbol{g}^T/B$, in (14) for two reasons: it is nonnegative definite and regularizes the inversion in (13); it vanishes when the parameters approach a stationary point. Actually, the Fisher information matrix could be singular for many commonly used models, e.g., finite mixture models, neural networks, hidden Markov models. We might not be able to inverse $\boldsymbol{F}$ for these singular statistical models without using regularization or damping. A Fisher type preconditioner with $\lambda > 0$ loses the scale-invariance property of a Newton type preconditioner. Both $\boldsymbol{P}$ and $\boldsymbol{P}^2$ can be useful preconditioners when the step size $\mu$ and damping factor $\lambda$ are set properly.

### 3.3 PROPERTIES OF THE NEWTON TYPE PRECONDITIONER

Following the ideas in (Li, 2018), we can show that (10) determines a unique positive definite preconditioner if and only if $E_v[\boldsymbol{v}\boldsymbol{v}^T]$ is positive definite and $\{E_v[\boldsymbol{v}\boldsymbol{v}^T]\}^{0.5}E_{z,v}[\hat{\boldsymbol{H}}\boldsymbol{v}\boldsymbol{v}^T\hat{\boldsymbol{H}}]\{E_v[\boldsymbol{v}\boldsymbol{v}^T]\}^{0.5}$ has distinct eigenvalues. Other minimum solutions of criterion (10) are either indefinite or negative definite, and are not interested for our purpose. The proof itself has limited novelty. We omit it here. Instead, let us consider the simplest case, where $\boldsymbol{\theta}$ is a scalar parameter, to gain some intuitive understandings of criterion (10). For scalar parameter, it is trivial to show that the optimal solutions minimizing (10) are

$$p = \pm\sqrt{E_v[v^2]/E_{z,v}[\hat{h}^2 v^2]} = \pm 1/\sqrt{h^2 + E_z[(h-\hat{h})^2]} \tag{15}$$

where $\hat{\boldsymbol{H}}, \boldsymbol{H}, \boldsymbol{P}$, and $\boldsymbol{v}$ are replaced with their plain lower case letters, and we have used the fact that $\boldsymbol{H} - \hat{\boldsymbol{H}}$ and $\boldsymbol{v}$ are independent. For gradient descent, we choose the positive solution, although the negative one gives the global minimum of (10). With the positive preconditioner, eigenvalue of the locally linear system in (5) is

$$h/\sqrt{h^2 + E_z[(h-\hat{h})^2]} \tag{16}$$

Now, it is clear that this optimal preconditioner damps the gradient noise when $E_z[(h-\hat{h})^2]$ is large, and preconditions the locally linear system in (5) such that its eigenvalue has unitary amplitude when the gradient noise vanishes. Convergence is ensured when a normalized step size, i.e., $0 < \mu < 1$, is used. For $\boldsymbol{\theta}$ with higher dimensions, eigenvalues of the locally linear system in (5) is normalized into range $[-1, 1]$ as well, in a way similar to (16).

## 4 ESTIMATING PRECONDITIONERS ON MATRIX LIE GROUPS

### 4.1 UPDATING RULE FOR THE NEWTON TYPE PRECONDITIONER

Let us take the Newton type preconditioner as an example to derive its updating rule. Updating rule for the Fisher type preconditioner is the same except for replacing the Hessian-vector product with stochastic gradient. Here, Lie group always refers to the matrix Lie group.

It is inconvenient to optimize $\boldsymbol{P}$ directly as it must be a symmetric and positive definite matrix. Instead, we represent the preconditioner as $\boldsymbol{P} = \boldsymbol{Q}^T\boldsymbol{Q}$, and estimate $\boldsymbol{Q}$. Now, $\boldsymbol{Q}$ must be a nonsingular matrix as both $c_n(\boldsymbol{P})$ and $c_f(\boldsymbol{P})$ diverge when $\boldsymbol{P}$ is singular. Invertible matrices with the same dimension form a Lie group. In practice, we are more interested in Lie groups with sparse representations. Examples of such groups are given in the next section. Let us consider a proper small perturbation of $\boldsymbol{Q}$, $\delta\boldsymbol{Q}$, such that $\boldsymbol{Q} + \delta\boldsymbol{Q}$ still lives on the same Lie group. The distance between $\boldsymbol{Q}$ and $\boldsymbol{Q} + \delta\boldsymbol{Q}$ can be naturally defined as $\mathrm{dist}(\boldsymbol{Q}, \boldsymbol{Q}+\delta\boldsymbol{Q}) = \sqrt{\mathrm{tr}(\delta\boldsymbol{Q}\boldsymbol{Q}^{-1}\boldsymbol{Q}^{-T}\delta\boldsymbol{Q}^T)}$ (Amari, 1998). Intuitively, this distance is larger for the same amount of perturbation when $\boldsymbol{Q}$ is closer to a singular matrix. With the above tensor metric definition, natural gradient for optimizing $\boldsymbol{Q}$ has form

$$\nabla\boldsymbol{Q} = \boldsymbol{R}\boldsymbol{Q} \tag{17}$$

For example, when $\boldsymbol{Q}$ lives on the group of invertible upper triangular matrices, $\boldsymbol{R}$ is given by

$$\boldsymbol{R} = 2\mathrm{triu}\{E_{z,v}[\boldsymbol{Q}\hat{\boldsymbol{H}}\boldsymbol{v}\boldsymbol{v}^T\hat{\boldsymbol{H}}^T\boldsymbol{Q}^T - \boldsymbol{Q}^{-T}\boldsymbol{v}\boldsymbol{v}^T\boldsymbol{Q}^{-1}]\} \tag{18}$$

where $\mathrm{triu}$ takes the upper triangular part of a matrix. Another way to derive (17) is to let $\delta\boldsymbol{Q} = \mathcal{E}\boldsymbol{Q}$, and consider the derivative with respect to $\mathcal{E}$, where $\mathcal{E}$ is a proper small matrix such that $\boldsymbol{Q}+\mathcal{E}\boldsymbol{Q}$ still lives on the same Lie group. Gradient derived in this way is known as relative gradient (Cardoso & Laheld, 1996). For our preconditioner learning problem, relative gradient and natural gradient have the same form. Now, $\boldsymbol{Q}$ can be updated using natural or relative gradient descent as

$$\boldsymbol{Q} \leftarrow \boldsymbol{Q} - \mu_q\boldsymbol{R}\boldsymbol{Q} \tag{19}$$

In practice, it is convenient to use the following updating rule with normalized step size,

$$\boldsymbol{Q} \leftarrow (\boldsymbol{I} - \mu_0\boldsymbol{R}/\|\boldsymbol{R}\|)\boldsymbol{Q} \tag{20}$$

where $0 < \mu_0 < 1$, and $\|\cdot\|$ takes the norm of a matrix. One simple choice for matrix norm is the maximum absolute value of a matrix.

Note that natural gradient can take different forms. One should not confuse the natural gradient on the Lie group derived from a tensor metric with the natural gradient for model parameter learning derived from a Fisher information metric.

## 4.2 SUMMARY OF THE NEWTON TYPE PRECONDITIONED SGD

One iteration of the Newton type preconditioned SGD consists of the following steps.

1. Evaluate stochastic gradient $\hat{\boldsymbol{g}}$.
2. Draw $\boldsymbol{v}$ from $\mathcal{N}(\boldsymbol{0}, \boldsymbol{I})$, and evaluate Hessian-vector product $\hat{\boldsymbol{H}}\boldsymbol{v}$.
3. Update preconditioners with $\boldsymbol{Q} \leftarrow (\boldsymbol{I} - \mu_0\hat{\boldsymbol{R}}/\|\hat{\boldsymbol{R}}\|)\boldsymbol{Q}$.
4. Update parameters with $\boldsymbol{\theta} \leftarrow \boldsymbol{\theta} - \mu\boldsymbol{Q}^T\boldsymbol{Q}\hat{\boldsymbol{g}}$.

The two step sizes, $\mu$ and $\mu_0$, are normalized. They should take values in range $[0, 1]$ with typical value $0.01$. We usually initialize $\boldsymbol{Q}$ to a scaled identity matrix with proper dimension. The specific form of $\hat{\boldsymbol{R}}$ depends on the Lie group to be considered. For example, for upper triangular $\boldsymbol{Q}$, we have $\hat{\boldsymbol{R}} = 2\mathrm{triu}[\boldsymbol{Q}\hat{\boldsymbol{H}}\boldsymbol{v}\boldsymbol{v}^T\hat{\boldsymbol{H}}^T\boldsymbol{Q}^T - \boldsymbol{Q}^{-T}\boldsymbol{v}\boldsymbol{v}^T\boldsymbol{Q}^{-1}]$, where $\boldsymbol{Q}^{-T}\boldsymbol{v}$ can be efficiently calculated with back substitution.

## 4.3 SUMMARY OF THE FISHER TYPE PRECONDITIONED SGD

We only need to replace $\hat{\boldsymbol{H}}\boldsymbol{v}$ in the Newton type preconditioned SGD with $\hat{\boldsymbol{g}}+\lambda\boldsymbol{v}$ to obtain the Fisher type one. Thus, we do not list its main steps here. Note that only its step size for the preconditioner updating is normalized. There is no simple way to jointly determine the proper ranges for step size $\mu$ and damping factor $\lambda$. Again, $\hat{\boldsymbol{R}}$ may take different forms on different Lie groups. For upper triangular $\boldsymbol{Q}$, we have $\hat{\boldsymbol{R}} = 2\mathrm{tiru}[\boldsymbol{Q}(\hat{\boldsymbol{g}} + \lambda\boldsymbol{v})(\hat{\boldsymbol{g}} + \lambda\boldsymbol{v})^T\boldsymbol{Q}^T - \boldsymbol{Q}^{-T}\boldsymbol{v}\boldsymbol{v}^T\boldsymbol{Q}^{-1}]$, where $\boldsymbol{v} \sim \mathcal{N}(\boldsymbol{0}, \boldsymbol{I})$. Here, it is important to note that the natural or relative gradient for $c_f(\boldsymbol{P})$ with the form given in (12) involves explicit matrix inversion. However, matrix inversion can be avoided by using the $c_f(\boldsymbol{P})$ in (11), which includes $\boldsymbol{v}$ as an auxiliary variable. It is highly recommended to avoid explicit matrix inversion for large $\boldsymbol{Q}$.

## 4.4 VARIATIONS OF PRECONDITIONED SGD

There are many ways to modify the above preconditioned SGD methods. Since curvatures typically evolves slower than gradients, one can update the preconditioner less frequently to save computations per iteration. With parallel computing available, one might update the preconditioner and model parameters simultaneously and asynchronously to save wall time per iteration. Combining preconditioner and momentum may further accelerate convergence. For recurrent neural network learning, we may need to clip the norm of preconditioned gradients to avoid excessively large parameter updates. In general, preconditioned gradient clipping relates to the trust region method by

$$\left\|\boldsymbol{\theta}^{[\mathrm{new}]} - \boldsymbol{\theta}\right\| = \|\mu\boldsymbol{P}\hat{\boldsymbol{g}}/\max(1, \|\boldsymbol{P}\hat{\boldsymbol{g}}\|/\Omega)\| \leq \mu\Omega$$

where $\Omega > 0$ is a clipping threshold, comparable to the size of trust region. One may set $\Omega$ to a positive number proportional to the square root of the number of model parameters. Most importantly, we can choose different Lie groups for estimating our preconditioners to achieve a good trade off between performance and complexity.

## 5 USEFUL LIE GROUPS WITH SPARSE REPRESENTATIONS

In practice, we seldom consider the Lie group consisting of dense invertible matrices for preconditioner estimation when the problem size is large. Lie groups with sparse structures are of more interests. To begin with, let us recall a few facts about Lie group. If $\boldsymbol{A}$ and $\boldsymbol{B}$ are two Lie groups, then $\boldsymbol{A}^T$, $\boldsymbol{A} \otimes \boldsymbol{B}$, and $\boldsymbol{A} \oplus \boldsymbol{B}$ all are Lie groups, where $\otimes$ and $\oplus$ denote Kronecker product and direct sum, respectively. Furthermore, for any matrix $\boldsymbol{C}$ with compatible dimensions, block matrix

$$\boldsymbol{Q} = \begin{bmatrix} \boldsymbol{A} & \boldsymbol{C} \\ \boldsymbol{0} & \boldsymbol{B} \end{bmatrix} \tag{21}$$

still forms a Lie group. We do not show proofs of the above statements here as they are no more than a few lines of algebraic operations. These simple rules can be used to design many useful Lie groups for constructing sparse preconditioners. We already know that invertible upper triangular matrices form a Lie group. Here, we list a few useful ones with sparse representations.

### 5.1 DIAGONAL PRECONDITIONER

Diagonal matrices with the same dimension and positive diagonal entries form a Lie group with reducible representation. Preconditioners learned on this group are called diagonal preconditioners.

### 5.2 KRONECKER PRODUCT PRECONDITIONER

For matrix parameter $\boldsymbol{\Theta}$, we can flatten $\boldsymbol{\Theta}$ into a vector, and precondition its gradient using a Kronecker product preconditioner with $\boldsymbol{Q}$ having form $\boldsymbol{Q} = \boldsymbol{Q}_2 \otimes \boldsymbol{Q}_1$. Clearly, $\boldsymbol{Q}$ is a Lie group as long as $\boldsymbol{Q}_1$ and $\boldsymbol{Q}_2$ are two Lie groups. Let us check its role in learning the following affine transformation

$$\boldsymbol{y} = [\boldsymbol{\Theta}_{\text{weight}}, \boldsymbol{\Theta}_{\text{bias}}]\boldsymbol{x} = \boldsymbol{\Theta}\boldsymbol{x} \tag{22}$$

where $\boldsymbol{x}$ is the input feature vector augmented with 1, and $\boldsymbol{y}$ is the output feature vector. After reverting the flattened $\boldsymbol{\Theta}$ back to its matrix form, the preconditioned SGD learning rule for $\boldsymbol{\Theta}$ is

$$\boldsymbol{\Theta} \leftarrow \boldsymbol{\Theta} - \mu \boldsymbol{Q}_1^T \boldsymbol{Q}_1 \frac{\partial \hat{f}(\boldsymbol{\Theta})}{\partial \boldsymbol{\Theta}} \boldsymbol{Q}_2^T \boldsymbol{Q}_2 \tag{23}$$

Similar to (3), we introduce coordinate transformation $\boldsymbol{\Theta}' = \boldsymbol{Q}_1^{-T}\boldsymbol{\Theta}\boldsymbol{Q}_2^{-1}$, and rewrite (23) as

$$\boldsymbol{\Theta}' \leftarrow \boldsymbol{\Theta}' - \mu \, \partial \hat{f}(\boldsymbol{\Theta}) / \partial \boldsymbol{\Theta}' \tag{24}$$

Correspondingly, the affine transformation in (22) is rewritten as $\boldsymbol{y}' = \boldsymbol{\Theta}'\boldsymbol{x}'$, where $\boldsymbol{y}' = \boldsymbol{Q}_1^{-T}\boldsymbol{y}$ and $\boldsymbol{x}' = \boldsymbol{Q}_2\boldsymbol{x}$ are the transformed input and output feature vectors, respectively. Hence, the preconditioned SGD in (23) is equivalent to the SGD in (24) with transformed feature vectors $\boldsymbol{x}'$ and $\boldsymbol{y}'$. We know that feature whitening and normalization could significantly accelerate convergence. A Kronecker product preconditioner plays a similar role in learning the affine transformation in (22).

### 5.3 SCALING AND NORMALIZATION PRECONDITIONER

This is a special Kronecker product preconditioner by constraining $\boldsymbol{Q}_1$ to be a diagonal matrix, and $\boldsymbol{Q}_2$ to be a sparse matrix where only its diagonal and last column can have nonzero values. Note that $\boldsymbol{Q}_2$ with nonzero diagonal entries forms a Lie group. Hence, $\boldsymbol{Q} = \boldsymbol{Q}_2 \otimes \boldsymbol{Q}_1$ is a Lie group as well. We call it a scaling and normalization preconditioner as it resembles a preconditioner that scales the output features and normalizes the input features. Let us check the transformed features $\boldsymbol{y}' = \boldsymbol{Q}_1^{-T}\boldsymbol{y}$ and $\boldsymbol{x}' = \boldsymbol{Q}_2\boldsymbol{x}$. It is clear that $\boldsymbol{y}'$ is an element-wisely scaled version of $\boldsymbol{y}$ as $\boldsymbol{Q}_1$ is a diagonal matrix. To make $\boldsymbol{x}'$ a "normalized" feature vector, $\boldsymbol{x}$ needs to be an input feature vector augmented with 1.

Let us check a simple example to verify this point. We consider an input vector with two features, and write down its normalized features explicitly as below,

$$
\begin{bmatrix} x_1' \\ x_2' \\ 1 \end{bmatrix} = \begin{bmatrix} 1/\sigma_1 & 0 & -m_1/\sigma_1 \\ 0 & 1/\sigma_2 & -m_2/\sigma_2 \\ 0 & 0 & 1 \end{bmatrix} \begin{bmatrix} x_1 \\ x_2 \\ 1 \end{bmatrix} \tag{25}
$$

where $m_i$ and $\sigma_i$ are the mean and standard deviation of $x_i$, respectively. It is straightforward to show that the feature normalization operation in (25) forms a Lie group with four freedoms. For the scaling-and-normalization preconditioner, we have no need to force the last diagonal entry of $\boldsymbol{Q}_2$ to be 1. Hence, the group of feature normalization operation is a subgroup of $\boldsymbol{Q}_2$.

### 5.4 SCALING AND WHITENING PRECONDITIONER

This is another special Kronecker product preconditioner by constraining $\boldsymbol{Q}_1$ to be a diagonal matrix, and $\boldsymbol{Q}_2$ to be an upper triangular matrix with positive diagonal entries. We call it a scaling-and-whitening preconditioner since it resembles a preconditioner that scales the output features and whitens the input features. Again, the input feature vector $\boldsymbol{x}$ must be augmented with 1 such that the whitening operation forms a Lie group represented by upper triangular matrices with 1 being its last diagonal entry. This is a subgroup of $\boldsymbol{Q}_2$ as we have no need to fix $\boldsymbol{Q}_2$'s last diagonal entry to 1.

It is not possible to enumerate all kinds of Lie groups suitable for constructing preconditioners. For example, Kronecker product preconditioner with form $\boldsymbol{Q} = \boldsymbol{Q}_3 \otimes \boldsymbol{Q}_2 \otimes \boldsymbol{Q}_1$ could be used for preconditioning gradients of a third order tensor. The normalization and whitening groups are just two special cases of the groups with the form shown in (21), and there are numerous more choices having sparsities between that of these two. Regardless of the detailed form of $\boldsymbol{Q}$, all such preconditioners share the same form of learning rule shown in (20), and they all can be efficiently learned using natural or relative gradient descent without much tuning effort.

## 6 RELATIONSHIP TO EXISTING METHODS

### 6.1 RELATIONSHIP TO ADAGRAD, RMSPROP AND ADAM

Adagrad, RMSProp and Adam all use the Fisher type preconditioner living on the group of diagonal matrices with positive diagonal entries. This is a simple group. Optimal solution for $c_f(\boldsymbol{P})$ has closed-form solution $\boldsymbol{P} = \mathrm{diag}(1 \oslash \sqrt{E_z[\hat{\boldsymbol{g}} \odot \hat{\boldsymbol{g}}] + \lambda^2})$, where $\odot$ and $\oslash$ denote element wise multiplication and division, respectively. In practice, simple exponential moving average is used to replace the expectation when using this preconditioner.

### 6.2 RELATIONSHIP TO EQUILIBRATED SGD

For diagonal preconditioner, the optimal solution minimizing $c_n(\boldsymbol{P})$ has closed-form solution $\boldsymbol{P} = \mathrm{diag}\left(\sqrt{E_v[\boldsymbol{v} \odot \boldsymbol{v}] \oslash E_{z,v}[\hat{\boldsymbol{H}}\boldsymbol{v} \odot \hat{\boldsymbol{H}}\boldsymbol{v}]}\right)$. For $\boldsymbol{v} \sim \mathcal{N}(\boldsymbol{0}, \boldsymbol{I})$, $E_v[\boldsymbol{v} \odot \boldsymbol{v}]$ reduces to a vector with unit entries. Then, this optimal solution gives the equilibration preconditioner in (Dauphin et al., 2015).

### 6.3 RELATIONSHIP TO KFAC AND SIMILAR METHODS

The preconditioners considered in (Povey et al., 2015) and (Martens & Grosse, 2015) are closely related to the Fisher type Kronecker product preconditioners. While KFAC approximates the Fisher metric of a matrix parameter as a Kronecker product to obtain its approximated inverse in closed-form solution, our method turns to an iterative solution to approximate this same inverse. Theoretically, our method's accuracy is only limited by the expressive power of the Lie group since no intermediate approximation is made. In practice, one distinct advantage of our method over KFAC is that explicit matrix inversion is avoided by introducing auxiliary vector $\boldsymbol{v}$ and using back substitution, while KFAC typically requires inversion of symmetric matrices. On graphics processing units (GPU) and with large matrices, parallel back substitution is as computationally cheap as matrix multiplication, and could be several orders of magnitude faster than inversion of symmetric matrix. Another advantage is that our method is derived from a unified framework. There is no need to invent different preconditioner learning rules when we switch the Lie group representations.

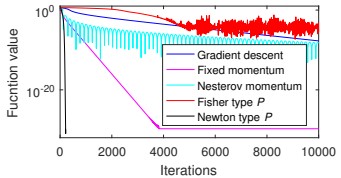

Figure 1: Convergence curves of compared methods on minimizing the Rosenbrock function.

## 6.4 RELATIONSHIP TO BATCH NORMALIZATION

Batch normalization can be viewed as preconditioned SGD using a specific scaling-and-normalization preconditioner with constraint $Q_1 = I$ and $Q_2$ from the feature normalization Lie group. However, we should be aware that explicit input feature normalization is only empirically shown to accelerate convergence, and has little meaning in certain scenarios, e.g., recurrent neural network learning where features may not have any stationary first or second order statistic. Both the Newton and Fisher type preconditioned SGD methods provide a more general and principled approach to find the optimal preconditioner, and apply to a broader range of applications. Generally, a scaling-and-normalization preconditioner does not necessarily "normalize" the input features in the sense of mean removal and variance normalization.

## 7 EXPERIMENTAL RESULTS

We use the square root Fisher type preconditioners in the following experiments since they are less picky on the damping factor, and seem to be more numerically robust on large scale problems. Still, as shown in our Pytorch implementation package, the original Fisher type preconditioners could perform better on small scale problems like the MNIST handwritten digit recognition task.

### 7.1 APPLICATION TO MATHEMATICAL OPTIMIZATION

Let us consider the minimization of Rosenbrock function, $f(\boldsymbol{\theta}) = 100(\theta_2 - \theta_1^2)^2 + (1 - \theta_1)^2$, starting from initial guess $\boldsymbol{\theta} = [-1, 1]$. This is a well known benchmark problem for mathematical optimization. The compared methods use fixed step size. For each method, the best step size is selected from sequence $\{\ldots, 1, 0.5, 0.2, 0.1, 0.05, 0.02, 0.01, \ldots\}$. For gradient descent, the best step size is 0.002. For momentum method, the moving average factor is 0.9, and the best step size is 0.002. For Nesterov momentum, the best step size is 0.001. For preconditioned SGD, $Q$ is initialized to $0.1I$ and lives on the group of triangular matrices. For the Fisher type method, we set $\lambda = 0.1$, and step sizes 0.01 and 0.001 for preconditioner and parameter updates, respectively. For the Newton type method, we set step sizes 0.2 and 0.5 for preconditioner and parameter updates, respectively. Figure 1 summarizes the results. The Newton type method performs the best, converging to the optimal solution using about 200 iterations. The Fisher type method does not fit into this problem, and performs poorly as expected. Mathematical optimization is not our focus. Still, this example shows that the Newton type preconditioned SGD works well for mathematical optimization.

### 7.2 IMAGENET EXPERIMENT

We consider the ImageNet ILSVRC2012 database for the image classification task. The well known AlexNet is considered. We follow the descriptions in (Alex et al., 2012) as closely as possible to set up our experiment. One main difference is that we do not augment the training data. Another big difference is that we use a modified local response normalization (LRN). The LRN function from TensorFlow implementation is not second order differentiable. We have to approximate the local energy used for LRN with a properly scaled global energy to facilitate Hessian-vector product evaluation. Note that convolution can be rewritten as correlation between the flattened input image patches and filter coefficients. In this way, we find that there are eight matrices to be optimized in the AlexNet, and their shapes are: $[96, 11 \times 11 \times 3 + 1], [256, 5 \times 5 \times 96 + 1], [384, 3 \times 3 \times 256 + 1], [384, 3 \times 3 \times 384 + 1], [256, 3 \times 3 \times 384], [4096, 6 \times 6 \times 256 + 1], [4096, 4096 + 1]$, and $[1000, 4096 + 1]$. We have tried diagonal and scaling-and-normalization preconditioners for

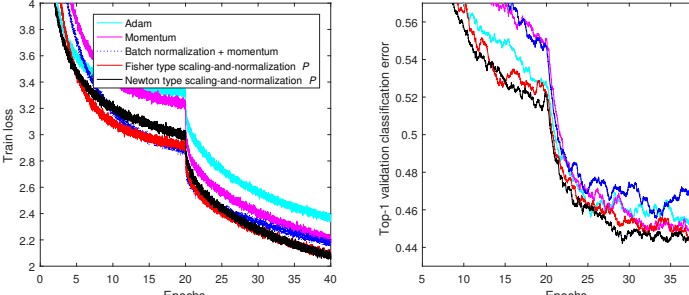

Figure 2: Typical smoothed learning curves from compared methods for the ImageNet ILSVRC2012 image classification task with AlexNet. Note that batch normalization alters the L2-regularization. Its training loss is not directly comparable with others.

each matrix. Denser preconditioners, e.g., the Kronecker product one, require hundreds of millions parameters for representations, and are expensive to run on our platform. Each compared method is trained with 40 epochs, mini-batch size 128, step size $\mu$ for the first 20 epochs, and $0.1\mu$ for the last 20 epochs. We have compared several methods with multiple settings, and only report the ones with reasonably good results here. For Adam, the initial step size is set to 0.00005. For batch normalization, initial step size is 0.002, and its moving average factors for momentum and statistics used for feature normalization are 0.9 and 0.99, respectively. The momentum method uses initial step size 0.002, and moving average factor 0.9 for momentum. Preconditioned SGD performs better with the scaling-and-normalization preconditioner. Its $Q$ is initialized to $0.1I$, and updated with normalized step size 0.01. For the Fisher type preconditioner, we set $\lambda = 0.001$ and initial step size 0.00005. For the Newton type preconditioner, its initial step size is 0.01. Figure 2 summarizes the results. Training loss for batch normalization is only for reference purpose as normalization alters the L2-regularization term. Batch normalization does not perform well under this setup, maybe due to its conflict with certain settings like the LRN and L2-regularization. We see that the scaling-and-normalization preconditioner does accelerate convergence, although it is super sparse. The Newton type preconditioned SGD performs the best, and achieves top-1 validation accuracy about 56% when using only one crop for testing, while the momentum method may require 90 epochs to achieve similar performance.

### 7.3 WORD LEVEL LANGUAGE MODELING EXPERIMENT

We consider the world level language modeling problem with reference implementation available from `https://github.com/pytorch/examples`. The Wikitext-2 database with 33278 tokens is considered. The task is to predict the next token from history observations. Our tested network consists of six layers, i.e., encoding layer, LSTM layer, dropout layer, LSTM layer, dropout layer, and decoding layer. For each LSTM layer, we put all its coefficients into a single matrix $\Theta$ by defining output and augmented input feature vectors as in $[i_t; f_t; g_t; o_t] = \Theta [x_t; h_{t-1}; 1]$, $c_t = f_t c_{t-1} + i_t g_t$, $h_t = o_t \tanh(c_t)$, where $t$ is a discrete time index, $x$ is the input, $h$ is the hidden state, and $c$ is the cell state. The encoding layer's weight matrix is the transpose of that of the decoding layer. Thus, we totally get three matrices to be optimized. With hidden layer size 200, shapes of these three matrices are $[4 \times 200, 2 \times 200 + 1]$, $[4 \times 200, 2 \times 200 + 1]$, and $[33278, 200 + 1]$, respectively. For all methods, the step size is reduced to one fourth of the current value whenever the current perplexity on validation set is larger than the best one ever found. For SGD, the initial step size is 20, and the gradient is clipped with threshold 0.25. The momentum method diverges easily without clipping. We set momentum 0.9, initial step size 1, and clipping threshold 0.25. We set initial step size 0.005 and damping factor $\lambda^2 = 10^{-12}$ for Adam and sparse Adam. Sparse Adam updates its moments and model parameters only when the corresponding stochastic gradients are not zeros. We have tried diagonal, scaling-and-normalization and scaling-and-whitening preconditioners for each matrix. The encoding (decoding) matrix is too large to consider KFAC like preconditioner. The diagonal preconditioner performs the worst, and the other two have comparable performance. For both types of preconditioned SGD, the clipping threshold for preconditioned gradient is 100, the initial step size is 0.1, and $Q$ is initialized to $I$. We set $\lambda = 0$ for the Fisher

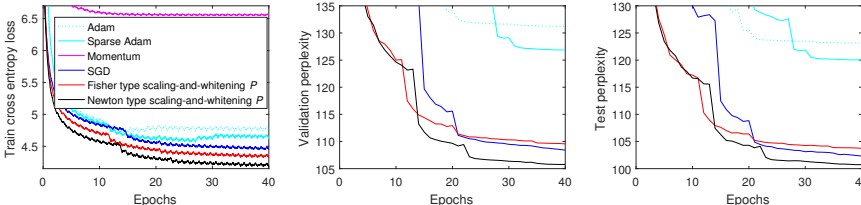

Figure 3: Typical learning curves from compared methods for the Wikitext-2 database word level language modeling task with a LSTM neural network.

type preconditioned SGD. With dropout rate $0.2$, the best three test perplexities are $100.7$ by SGD, $103.9$ by Newton type preconditioned SGD, and $105.8$ by Fisher type preconditioned SGD. With dropout rate $0.35$, the best three test perplexities are $100.7$ by Newton type preconditioned SGD, $102.3$ by SGD, and $103.7$ by Fisher type preconditioned SGD. Although SGD performs well on the test set, both types of preconditioned SGD have significantly lower training losses than SGD with either dropout rate. Figure 3 summarizes the results when the dropout rate is $0.35$. Methods involving momentum, including Adam and sparse Adam, perform poorly. Note that our preconditioners preserve the sparsity property of gradients from the encoding and decoding layers (Appendix A). This saves considerable computations by avoiding update parameters with zero gradients. Again, both preconditioners accelerate convergence significantly despite their high sparsity.

### 7.4 COMPUTATIONAL COMPLEXITY AND IMPLEMENTATION

Compared with SGD, the Fisher type preconditioned SGD adds limited computational complexity when sparse preconditioners are adopted. The Newton type preconditioned SGD requires Hessian-vector product, which typically has complexity comparable to that of gradient evaluation. Thus, using SGD as the base line, the Newton type preconditioned SGD approximately doubles the computational complexity per iteration, while the Fisher type SGD has similar complexity. Wall time per iteration of preconditioned SGD highly depends on the implementations. Ideally, the preconditioners and parameters could be updated in a parallel and asynchronous way such that SGD and preconditioned SGD have comparable wall time per iteration.

We have put our TensorFlow and Pytorch implementations on `https://github.com/lixilinx`. More experimental results comparing different preconditioners and optimization methods on diverse benchmark problems can be found there. For the ImageNet experiment, all compared methods are implemented in Tensorflow, and require two days and a few hours to finish 40 epochs on a GeForce GTX 1080 Ti GPU. The word level language modeling experiment is implemented in Pytorch. We have rewritten the word embedding function to enable second order derivative. For this task, SGD and the Fisher type preconditioned SGD have similar wall time per iteration, while the Newton type method requires about $80\%$ more wall time per iteration than SGD when running on the same GPU.

## 8 CONCLUSIONS

Two types of preconditioners and preconditioned SGD methods are studied. The one requiring Hessian-vector product for preconditioner estimation is suitable for general purpose optimization. We call it the Newton type preconditioned SGD due to its close relationship to the Newton method. The other one only requires gradient for preconditioner estimation. We call it the Fisher type preconditioned SGD as its preconditioner is closely related to the inverse of Fisher information matrix. Both preconditioners can be efficiently learned using natural or relative gradient descent on any matrix Lie groups designated by the user. The Fisher type preconditioned SGD has lower computational complexity per iteration, but may require more tuning efforts on selecting its step size and damping factor. The Newton type preconditioned SGD has higher computational complexity per iteration, but is more user friendly due to its use of normalized step size and built-in gradient noise damping ability. Both preconditioners, even with very sparse representations, are shown to considerably accelerate convergence on relatively large scale problems.

ACKNOWLEDGMENTS

The author thanks the reviewers for their comments and Yaroslav Bulatov for his discussions to improve this paper.

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

## APPENDIX A: WHITENING-AND-SCALING PRECONDITIONER PRESERVES SPARSITY OF GRADIENT OF ENCODING MATRIX

We are to show that the scaling-and-whitening and whitening-and-scaling preconditioners preserve the sparsity property of gradients from the decoding and encoding layers in the word level language model considered in Experiment 3, respectively. Clearly, we only need to show this for the encoding part. Let

$$\boldsymbol{W} = [\boldsymbol{w}_1, \boldsymbol{w}_2, \ldots, \boldsymbol{w}_I]$$

be a $D \times I$ word embedding matrix, where $D$ is the embedding dimension, $I$ is the number of tokens, and $\boldsymbol{w}_i$ is the vector representation for the $i$th token. Typically, only a whitening-and-scaling and sparser preconditioners are affordable since $I \gg D > 1$. Notably, for sufficiently large $I$, the whitening-and-scaling preconditioner could be sparser than the diagonal one. Let us consider preconditioner

$$\boldsymbol{P} = \boldsymbol{Q}^T\boldsymbol{Q}, \quad \boldsymbol{Q} = \boldsymbol{Q}_2 \otimes \boldsymbol{Q}_1, \quad \boldsymbol{Q}_2 = \mathrm{diag}(q_{2,1}, q_{2,2}, \ldots, q_{2,I})$$

By (23), the preconditioned gradient is

$$[q_{2,1}^2\boldsymbol{Q}_1^T\boldsymbol{Q}_1\hat{\boldsymbol{g}}_1, q_{2,2}^2\boldsymbol{Q}_1^T\boldsymbol{Q}_1\hat{\boldsymbol{g}}_2, \ldots, q_{2,I}^2\boldsymbol{Q}_1^T\boldsymbol{Q}_1\hat{\boldsymbol{g}}_I]$$

where $\hat{\boldsymbol{g}}_i$ is the stochastic gradient for the $i$th word embedding vector. Since $I \gg B$, most of these $\hat{\boldsymbol{g}}_i$'s are zeros, where $B$ is the batch size. This preconditioner does not mix up gradients of different word embedding vectors. Hence, the sparsity property is preserved.

