# OpenReview forum: "Preconditioner on Matrix Lie Group for SGD"
_ICLR.cc/2019/Conference_

### Official Review · AnonReviewer2 · 2018-10-16
**Solid work**

**Rating:** 7
**Confidence:** 3

**Review:**

The authors suggest and analyse two types of preconditioners for optimization, a Newton type and a Fisher type preconditioner.

The paper is well written, the analysis is clear and the significance is arguably given. The authors run their optimizers on a synthetic benchmark data set and on imagenet.
The originality is not so high as the this line of research exists for long.
The "Lie" in the title is (technically correct, but) a bit misleading, as only matrix groups were used.

---

> ### Author Response · Authors · 2018-11-07
> **Responses to AnonReviewer2’s comments**
>
> [Comment 1]: The "Lie" in the title is (technically correct, but) a bit misleading, as only matrix groups were used.
> [Response 1]: We will use ‘matrix Lie group’ in the title after revision. In the text, we already point out that Lie group in the paper refers to the matrix Lie group.

---

### Official Review · AnonReviewer1 · 2018-11-04
**Possibly interesting ideas but badly presented and justified, and with poor experimental design**

**Rating:** 5
**Confidence:** 5

**Review:**

This paper proposes a preconditioned SGD method where the preconditioner is adapted by performing some type of gradient descent on some secondary objective "c".  The preconditioner lives in one of a restricted class of invertible matrices (e.g. symmetric, diagonal, Kronecker-factored) constituting a Lie group (which is where the title comes from).

I think the idea of designing a preconditioner based on considerations of gradient noise and as well as the Hessian is interesting. However most of that work was done in the Li paper, and including the design of "c".  This paper's contribution seems to be to work out some of the details for various restricted classes of matrices, to construct a "Fisher version" of c, and to run some experiments.

The problem is that I don't really buy the original motivation for the "c" function from the Li paper, and the newer Fisher version of c proposed in this paper doesn't seem to have any justification at all.  I also find that the paper in general doesn't do a good job of explaining its various choices when designing the algorithm.  This could be somewhat forgiven if the experimental results were strong, but unfortunately they are too limited, and marred by overly-simplistic baselines that aren't properly tuned.


More detailed comments below

Title:

I think the title is poorly chosen.  The paper doesn't use Lie groups or their properties in any significant way, and "learning" is a bad choice of words too, since it involves generalization etc (it's not merely the optimization of some function).  A better title would be "A general framework for adaptive preconditioners" or something.

Intro:

Citation of Adagrad paper is broken

The literature review contained in the intro needs works. I wouldn't call methods like quasi-Newton methods "convex optimization methods".  Those algorithms were around a long time ago before "convex optimization" was a specific topic of study and are probably *less* associated with the convex optimization literature than, say, Adagrad is. And methods like Adagrad aren't exactly first-order methods either. They use adaptively chosen preconditioners (that happen to be diagonal) which puts them in a similar category to methods like LBFGS, KFAC, etc.

It's not clear at this point in the paper what it means for a preconditioner to be "learned on" something.

Section 2:

The way you discuss quadratic approximations is confusing.  Especially  the sentence "is the sum of approximation error and constant term independent of theta" where you then go on to say that a_z does depend on theta.  I know that this second theta is the "current theta" separate from the theta as it appears in the formula for the approximation but this is really sloppy. Usually people construct the quadratic approximation in terms of the *change in theta* which makes such things cleaner.

You should explain how eqn 8 was derived since it's so crucial to everything that follows.  Citing a previous paper with no further explanation really isn't good enough here.  Surely with all of the notation you have already set up it should be possible to motivate this criterion somehow. The simple fact that it recovers P = H^-1 in the noiseless quadratic case isn't really good enough, since many possible criteria would do the same.

I've skimmed the paper you cited and their justification for this criterion isn't very convincing.  There are other possible criteria that they give and there doesn't seem to be a strong reason to prefer one over the other.


Section 3:

The way you define the Fisher information matrix corresponds to the "empirical Fisher", since z includes the training labels.  This is different from the standard Fisher information matrix.

How can you motivate doing the "replacement" that you do to generate eqn 12? Replacing delta theta with v is just notation, but how can you justify replacement of delta g with g + lambda v?  This isn't a reasonable approximation in any sense that I can discern. Once again this is an absolutely crucial step that comes out of nowhere.  Honestly it feels contrived in order to produce a connection to popular methods like Adam.

Section 4:

The prominent use of the abstract mathematical term "Lie group" feels unnecessary and like mathematical name-dropping. Why not just talk about certain "classes" of invertible matrices closed under standard operations (which would also help people that don't know what a Lie group is)?  If you are going to invoke some abstract mathematical framework like Lie groups it needs to actually help you do something you couldn't otherwise. You need to use some kind of advanced Theorem for Lie groups.

Without knowing the general form of R equation 18 is basically vacuous. *any* matrix (in the same class) could be written this way.

I've never heard of the natural gradient being defined using a different metric than the Fisher metric.  If the metric can be arbitrary then even standard gradient descent is a "natural gradient" too (taking the Euclidean metric).  You could argue for a generalized definition that would include only parametrization independent metrics, but then your particular metric wouldn't obviously work.


Section 6:

Rather than comparing to Batch Normalization you would be better off comparing to the old centering and normalization work of Schraudolph et al which the former was based on, which is actually a well-defined preconditioner.

Section 7:

You really need to sweep over the learning rate parameters for optimizers like SGD with momentum or Adam.   Otherwise the comparisons aren't very interesting.

"Tikhonov regularization" should just be called L2-regularization

---

> ### Author Response · Authors · 2018-11-08
> **Responses to AnonReviewer1’s comments**
>
> [Comment 1]: …most of that work was done in the Li paper…
> [Response 1]: Our contributions include: propose a new framework for learning preconditioners on Lie groups; predict useful new preconditioners and optimization methods; reveal its relationships to many existing methods (ESGD, batch normalization, KFAC, Adam, RMSProp, Adagrad); compare Newton and Fisher type preconditioners; implementations and empirical performance study.
>
> [Comment 2]: … I don't really buy the original motivation for the "c" function…doesn't seem to have any justification at all…
> [Response 2]: We are willing to know the reasons.
>
> [Comment 3]: …experimental results…too limited…overly-simplistic baselines…aren't properly tuned…
> [Response 3]: You may find more comparison results on small scale problems like MNIST related in our implementation packages. The image recognition and NLP tasks considered in the paper are representative, and baselines already achieved reasonable performance. Still, we are willing to fine tune them and update the results during the rebuttal period.
>
> [Comment 4]: …title is poorly chosen…doesn't use Lie groups…in any significant way… "learning" is a bad choice of words…merely the optimization of some function…
> [Response 4]: Solving for the optimal preconditioner is a tracking problem since generally the Hessian changes along with parameters, and also an estimation problem due to the existence of gradient noises. So we think ‘learning’ is a proper word. Lie group provides a concise framework for our study, and enables efficient learning via natural gradient descent.
>
> [Comment 5]: I wouldn't call methods like quasi-Newton methods "convex optimization methods"...less associated with the convex optimization literature…
> [Response 5]: Quasi-Newton methods are derived assuming nonnegative definite Hessian, and are taught in convex optimization textbooks.
>
> [Comment 6]: Citation of Adagrad paper is broken... methods like Adagrad aren't exactly first-order methods...
> [Response 6]: We state that Adagrad is a variation of SGD. We do not state that it is a first-order method.
>
> [Comment 7]: The way you discuss quadratic approximations is confusing...a_z...really sloppy...people construct the quadratic approximation in terms of the *change in theta*...
> [Response 7]: We will explicitly point out that a_z only contains higher order approximation errors in the revised paper.
> You can construct quadratic approximation in terms of either theta or the change in theta.
>
> [Comment 8]: You should explain how eqn 8 was derived...Citing a previous paper...isn't good enough…I've skimmed the paper you cited... justification for this criterion isn't very convincing...
> [Response 8]: We believe these topics are thoroughly addressed in the cited paper. This is a conference paper with recommend page length 8. Nevertheless, we reviewed important facts in the background section, e.g., Eq. (9), the correspondence to Newton method regardless of the existence of nonconvexity and gradient noises.
>
> [Comment 9]: The way you define the Fisher information matrix corresponds to the "empirical Fisher"...
> [Response 9]: We will emphasize it in the revised paper. We already emphasized it in our implementations.
>
> [Comment 10]: How can you motivate doing the "replacement" …how can you justify replacement of …  This isn't a reasonable approximation in any sense... comes out of nowhere…it feels contrived to…
> [Response 10]: The math in the paper is clear. No approximation is involved here.
>
> [Comment 11]: … use of the abstract mathematical term "Lie group" feels unnecessary…mathematical name-dropping…Why not … "classes" of invertible matrices closed under standard operations…You need to use some kind of advanced Theorem for Lie groups.
> [Response 11]: Matrix Lie group is the precise term here. We use its properties to design the preconditioners and their learning rules.
>
> [Comment 12]: … equation 18 is basically vacuous…
> [Response 12]: It is Amari’s natural gradient or Cardoso’s relative gradient on the Lie group.
>
> [Comment 13]: I've never heard of the natural gradient being defined using a different metric than the Fisher metric…your particular metric wouldn't obviously work.
> [Response 13]: Please check Amari’s work on natural gradient.
>
> [Comment 14]: Rather than comparing to Batch Normalization you would be better off comparing to the old centering and normalization work of Schraudolph…
> [Response 14]: Please give further details like Schraudolph’s paper, link, code implementation, etc.
>
> [Comments 15]: You really need to sweep over the learning rate parameters for optimizers like SGD...
> [Response 15]: We already searched the learning rates in a large range for these methods. We are further refining the results of SGD, momentum and Adam, and update the paper during the rebuttal period.
>
> [Comment 16]: "Tikhonov regularization" should just be called L2-regularization
> [Response 16]: We will call it L2-regularization in the revised paper.

---

### Official Review · AnonReviewer3 · 2018-11-06
**explanation could use more work, but a solid idea that seems to work in practice**

**Rating:** 8
**Confidence:** 5

**Review:**

Author proposes general framework to use gradient descent to learn a preconditioner related to inverse of the Hessian, or the inverse of Fisher Information matrix, where the inverse may take a particular form, ie, Kronecker-factored form like in KFAC. I have tracked down the implementation of this method by author from earlier paper Li 2018 and verified that it works and speeds up convergence of convolutional networks in terms of number of iterations needed. In particular, Kronecker Factored preconditioner using approach in the paper worked better in terms of wall-clock time on MNIST LeNet5, comparing against an existing PyTorch implementation of KFAC from César Laurent.


Some comments on the paper:

Section 2
The key seems to be equation 8. The author provides loss function, the minimum is what is achieved by inverse of the Hessian. Given the importance of the formula, it feels like proof should be included (perhaps in Appendix).

Justification of the criterion is relegated to earlier work in Li (https://arxiv.org/pdf/1512.04202.pdf), but I failed to fully grasp the motivation. There are simpler criteria being introduced, such as criterion 1, equation 17, which simply minimizes the difference between predicted gradient delta and observed, why not use that criterion?

The justification is given that using inverse Hessian may "amplify noise", which I don't buy. When using SGD to solve least-square regression, dividing by Hessian does not have a problem of amplifying noise, so why is this a concern here?


Section 3

The paper should make it clear that empirical Fisher matrix is used, unlike "unbiased estimate of true Fisher" which used in many natural gradient papers.

Section 4
Is "Lie group" used anywhere in the derivations? It seems the same algebra holds even without that assumption. The motivation for using "natural gradient for learning Q" seems to come from Amari. I have not read that paper, how important it is to use the "natural" gradient for learning Q? What if we use regular gradient descent for Q?

Section 7
Figure 1 showed that Fisher-type criterion didn't work for toy problem, it would be more informative if it used square root of Fisher-type criterion. The square root comes out of regret-analysis (ie, AdaGrad uses square root of gradient covariance)

---

> ### Author Response · Authors · 2018-11-07
> **Responses to AnonReviewer3’s comments**
>
> [Comment 1]: The key seems to be equation 8. ...it feels like proof should be included...
> [Response 1]: We believe this equation is thoroughly studied in Li’s work.
>
>
> [Comment 2]: Justification of the criterion
> [Response 2]: Let us consider three cases to compare these criteria.
>
> Case 1, noiseless gradient, positive definite Hessian. All preconditioners in Li’s work are equivalent, leading to the same secant equation, delta g = H * delta x.
>
> Case 2, noiseless gradient, indefinite Hessian. Only criterion c3 in Li’s work can guarantee positive definiteness of the preconditioner. One may point out that other criteria can yield positive definite preconditioner under Wolfe conditions. But the resultant preconditioner is remotely related to the Hessian. We are seeking a preconditioner whose eigenvalues are the inverse of the absolute eigenvalues of Hessian to precondition the Hessian perfectly.
>
> Case 3, noisy gradient, positive definite or indefinite Hessian. Only criterion c3 in Li’s work leads to a preconditioner that still corresponds to the secant equation delta g = H * delta x for math optimization, i.e., eq. (9) shown in our paper.
>
>
> [Comment 3] The justification is given that using inverse Hessian may "amplify noise", which I don't buy...
> [Response 3]: Gradient noise amplification may cause little concern for well-conditioned problems. Your least-square regression problem might fall into this case. But it could lead to divergence for ill-conditioned problems, e.g., learning recurrent networks requiring long term memory. Fig. 6 in Li’s work shows one such example.
>
> Using SGD as the base line, a good preconditioner actually suppresses the gradient noise. Our Tensorflow implementation gives one RNN training example using batch size 1. SGD fails to converge with batch size 1, although it converges with much large batch sizes. Our methods converge well with batch size 1 since the preconditioners also suppress gradient noises.
>
>
> [Comment 4]: The paper should make it clear that empirical Fisher matrix is used...
> [Response 4]: We will emphasize it in the revised paper. We already emphasized it in our implementation packages.
>
>
> [Comment 5]: Is "Lie group" used anywhere in the derivations...
> [Response 5]: These are great questions. We choose to learn the preconditioner on Lie group due to our years of practices in neural network training. Properties of Lie group are repeatedly exploited by our methods. For example, you mentioned that ‘same algebra holds even without that assumption’. Well, it is true because Q and Q + delta Q are already on the same Lie group. Otherwise, this is not necessarily true. For example, if you constrain Q to be a band matrix, generally, you may not able to write delta Q as -(step size)*R*Q, where R is a band matrix similar to Q.
>
> Why natural gradient? Once we decide to learn the preconditioner on the Lie group, then gradient on the Lie group is just the natural gradient derived from a tensor metric. On the theoretical aspects, both Amari and Cardoso give a lot of justifications for natural gradient, i.e., equivariant property, fast convergence, etc. In practice, it helps a lot as we can use normalized step size to update the preconditioner. We rarely feel the need to tune this step size (0.01 as default value and works well).
>
> Can we use regular gradient descent? Let us consider two cases.
>
> Case 1, Q is on a Lie group. Yes, we can use regular gradient descent. But the updating step size may require fine tuning for each specific problem. Convergence could be slow when initial values for Q is either too large or too small. Precautions are required to prevent Q converging to singular matrices.
>
> Case 2, Q is not on any Lie group. Regular gradient descent still works. Similar difficulties are: how to choose the updating step size; how to determine the initial value. For example, the authors have considered preconditioner with form P = (scalar)*I + U*U^T. For math optimization, we already know how to update this preconditioner (limited-memory BFGS). For stochastic optimization, the authors failed to find an efficient and yet tuning-free updating methods for such preconditioner. However, we do not exclude the existence of such preconditioner updating methods.
>
>
> [Comment 6]: ...it would be more informative if it used square root of Fisher-type criterion...
> [Response 6]: We used square root Fisher-type preconditioner. We will clarify it in the revised paper.
>
> By the way, our Pytorch implementation gives demo showing the usage of both square root and regular Fisher type preconditioners. For small scale problems like MNIST, the Fisher type preconditioner may perform better. For large scale problems, the square root Fisher type preconditioner seems more numerically robust, less picky on the damping factor. So we use the square root Fisher type preconditioner in experiment 2 and 3.

---

### Author Response · Authors · 2018-11-06
**natural gradient can be derived from different metrics**

Just a quick response to AnonReviewer1's comment stating that 'I've never heard of the natural gradient being defined using a different metric than the Fisher metric'. This is not true. Please check Amari's classic paper, Natural Gradient Works Efﬁciently in Learning, sections 3.3, 3.4, 7, and 8, for examples of natural gradient on Lie groups.

Actually, considering that some readers might not be familiar with natural gradient, we have a note at the end of section 4.1 of our paper to remind the difference between a natural gradient derived from the Fisher metric and a natural gradient derived from a tensor metric.

We thank reviewers for their time and efforts, and will improve our paper accordingly.

---

### Author Response · Authors · 2018-11-13
**List of revisions**

1, Section 4.4, add a note on preconditioned gradient norm clipping and its relationship to trust region method due to its importance in practice.

2, Section 6.3, add a few comments on complexity comparison with KFAC. KFAC requires inversion of symmetric matrices, and thus might fail to scale up to large scale problems since generally, it is difficult to efficiently inverse a matrix in parallel (to our knowledge and experiences). Our methods require back substitution, which is as computationally cheap as matrix multiplication on GPU (given enough resources for parallelization). Thus, our methods could scale up to large scale problems.

Please check code (in our pytorch implementation, misc dir)
https://github.com/lixilinx/psgd_torch/blob/master/misc/benchmark_mm_trtrs_inv.py
for details. For linear system with dimension 1024, back substitution is about 300 times faster than matrix inversion on 1080 ti GPU. For dimension 8192, back substitution is about 2000 times faster.

3, Sections 7.2 and 7.3, fine tune performance of SGD, momentum and Adam, especially on the language modeling task since momentum and Adam perform poorly due to the sparsity of gradients. For this task, we found that:

Momentum: diverge when step size > =0.2; converges when step size <= 0.05. Convergence is too slow. Then we tried to clip the updates as in clipped SGD method to avoid divergence when large step size is used. Still, momentum method performs the worst.

Adam: Fine tune its damping factor improves performance. Its performance is rather sensitive to the damping factor. As the momentum method, Adam also destroys the sparsity of gradients and performs not so well.

Sparse Adam: it only updates the 1st and 2nd moments and model parameters when their corresponding gradients are not zeros. This slightly improves performance, but still far from SGD’s performance.

One may argue that SGD is a special case of momentum and Adam, and thus they should perform as well as SGD after fine tuning all their parameters. Well, we do not agree for two reasons. First, jointly fine tuning all these parameters are too expensive. For example, Adam has four parameters to tweak. Second, as their names suggest, certain parameters are expected to have their typical values. For example, a momentum method with momentum 0 is just SGD, but not a typical momentum method.

4, Appendix A, a short note showing that our preconditioners preserve the sparsity of gradients in the language modeling task.

5, Scattered minor revisions and clarifications in the text. Many are due to reviews’ comments, and we appreciate it.

---

### Meta-Review · Area_Chair1 · 2018-12-08
**clever idea, but experiments and writing need much improvement**

**Confidence:** 3
**Recommendation:** Accept (Poster)

**Metareview:**

The method presented here adapts an SGD preconditioner by minimizing particular cost functions which are minimized by the inverse Hessian or inverse Fisher matrix. These cost functions are minimized using natural (or relative) gradient on the Lie group, as previously introduced by Amari. This can be extended to learn a Kronecker-factored preconditioner similar to K-FAC, except that the preconditioner is constrained to be upper triangular, which allows the relative gradient to be computed using backsubstitution rather than inversion. Experiments show modest speedups compared to SGD on ImageNet and language modeling.

There's a wide divergence in reviewer scores. We can disregard the extremely short review by R2. R1 and R3 each did very careful reviews (R3 even tried out the algorithm), but gave scores of 5 and 8. They agree on most of the particulars, but just emphasized different factors. Because of this, I took a careful look, and indeed I think the paper has significant strengths and weaknesses.

The main strength is the novelty of the approach. Combining relative gradient with upper triangular preconditioners is clever, and allows for a K-FAC-like algorithm which avoids matrix inversion. I haven't seen anything similar, and this method seems potentially useful. R3 reports that (s)he tried out the algorithm and found it to work well. Contrary to R1, I think the paper does use Lie groups in a meaningful way.

Unfortunately, the writing is below the standards of an ICLR paper. The title is misleading, since the method isn't learning a preconditioner "on" the Lie group. The abstract and introduction don't give a clear idea of what the paper is about. While some motivation for the algorithms is given, it's expressed very tersely, and in a way that will only make sense to someone who knows the mathematical toolbox well enough to appreciate why the algorithm makes sense. As the reviewers point out, important details (such as hyperparameter tuning schemes) are left out of the experiments section.

The experiments are also somewhat problematic, as pointed out by R1. The paper compares only to SGD and Adam, even though many other second-order optimizers have been proposed (and often with code available). It's unclear how well the baselines were tuned, and at the end of the day, the performance gain is rather limited. The experiments measure only iterations, not wall clock time.

On the plus side, the experiments include ImageNet, which is ambitious by the standards of an algorithmic paper, and as mentioned above, R3 got good results from the method.

On the whole, I would favor acceptance because of the novelty and potential usefulness of the approach. This would be a pretty solid submission of the writing were improved. (While the authors feel constrained by the 8 page limit, I'd recommend going beyond this for clarity.) However, I emphasize that it is very important to clean up the writing.

---

> ### Author Response · Authors · 2018-12-24
> **Thanks for the clear and fair review, just a few comments clarifying the experimental parts**
>
> COMPARISON WITH SECOND-ORDER METHODS:
>
> Our Tensorflow/Pytorch implementations give such comparisons on small scale problems like the MNIST dataset, which suggest that our methods are competitive, and the Newton version might give slightly better train/test errors than others.  For large scale problems, many second-order methods including KFAC could become so computationally expensive that massive parallel computing is required for efficient implementations. Furthermore, regarding several KFAC’s open source implementations, to our knowledge, it is unclear on how to handle certain common designs like tied weights (experiment 3), L1/L2 regularizations (experiment 2; the KFAC paper does discuss L2 regularizations, but some implementations mix up regularization and damping), dropout (experiments 2&3), etc.
>
> For example, in AlexNet, the largest matrix has shape [4096, 9217].  Just for this single matrix, KFAC needs about 100M coefficients saving its Kronecker-product-factorized Fisher matrices, and inverting a 9217x9217 matrix repeatedly, which could take seconds, depending the hardware. In the language modeling example, the largest matrix has shape [33278, 201]. KFAC needs over one billion coefficients saving the Fisher! On the other hand, our methods provide the choices to use very sparse preconditioners, and may require much less computing resources. For example, for the largest matrices, our preconditioner only has about 22K coefficients in the AlexNet, and about 52K coefficients in the language modeling example.
>
> HYPERPARAMETER TUNING:
>
> Hyperparameters like batch size, learning rate annealing scheme, network structure and sizes, dropout rate, L1/L2 regularization, number of epochs, etc., adopt their typical values given in either previous papers or popular implementations. Nevertheless, for the AlexNet, we only take 40 epochs, instead of 90 epochs, to save experimental time.
>
> For the Newton type preconditioned SGD, both step sizes for parameter learning and preconditioner adaptation are normalized, and value 0.01 is a good guess. If aggressive gradient clipping is used (a common practice in training RNNs), we may increase the step size for parameter learning to 0.1. We do not tune these two step sizes, and simply use these default values in experiments 2 and 3.
>
> The Fisher type preconditioned SGD and many other methods like momentum, Adam, etc., do not use normalized step size and involve more hyperparameters. We mainly search their learning rates in a large range, and keep other parameters their default values. Still we found that their performance could be sensitive to certain hyperparameters like damping factor, and their tweakings are necessary as well.
>
> PERFORMANCE GAIN/WALL TIME COMPARISON:
>
> On small scale problems, large performance gains over baselines like SGD can be achieved by using denser preconditioners. On large scale problems as considered in the paper, the gain may be limited, but achieved with limited cost due to the use of extremely sparse preconditioner (sparser than the diagonal preconditioner). Wall time per iteration depends on the implementation and hardware. On a single GPU, for the AlexNet example, wall time of our methods is comparable to that of SGD (less than 10% difference); for the language modeling example, wall time of our Fisher type method is comparable to that of SGD, while the Newton type version takes about 1.8 times wall time of SGD.